# Measuring Privacy Risks and Tradeoffs in Financial Synthetic Data Generation

### Michael Zuo
Rensselaer Polytechnic Institute
Troy, New York, USA
zuom@rpi.edu

### Inwon Kang
Rensselaer Polytechnic Institute
Troy, New York, USA
kangi@rpi.edu

### Stacy Patterson
Rensselaer Polytechnic Institute
Troy, New York, USA
sep@cs.rpi.edu

### Oshani Seneviratne
Rensselaer Polytechnic Institute
Troy, New York, USA
senevo@rpi.edu

## Abstract

We explore the privacy-utility tradeoff of synthetic data generation schemes on tabular financial datasets, a domain characterized by high regulatory risk and severe class imbalance. We consider representative tabular data generators, including autoencoders, generative adversarial networks, diffusion, and copula synthesizers. To address the challenges of the financial domain, we provide novel privacy-preserving implementations of GAN and autoencoder synthesizers. We evaluate whether and how well the generators simultaneously achieve data quality, downstream utility, and privacy, with comparison across balanced and imbalanced input datasets. Our results offer insight into the distinct challenges of generating synthetic data from datasets that exhibit severe class imbalance and mixed-type attributes.

## CCS Concepts

• **Security and privacy** → **Data anonymization and sanitization**; *Privacy protections*; • **Computing methodologies** → *Machine learning*.

## Keywords

Differential Privacy, Membership Inference Attack, Tabular Data, Privacy Audit, Synthetic Data Generation

**ACM Reference Format:**
Michael Zuo, Inwon Kang, Stacy Patterson, and Oshani Seneviratne. 2026. Measuring Privacy Risks and Tradeoffs in Financial Synthetic Data Generation. In *Proceedings of The ACM Web Conference'26 (The Web Conference '26)*. ACM, New York, NY, USA, 11 pages. https://doi.org/XXXXXXX.XXXXXXX

## 1 Introduction

Machine learning underpins many of today's web-scale applications, from recommendation and personalization systems to fraud detection and e-commerce analytics that rely on large volumes of user data. The success of such models is heavily reliant on the availability of high-quality data. However, regulations like the General Data Protection Regulation (GDPR) [12] or the California Consumer Privacy Act (CCPA) [18] impose strict limitations on how customer data can be used, shared, and analyzed. These constraints are particularly acute in domains involving sensitive user attributes, such as financial transactions, browsing behavior, or health records, where the risk of deanonymization or misuse threatens both individuals and platforms.

Synthetic data generation [7] offers a promising solution to this dilemma. By training a generative model on a private dataset, an organization can produce an artificial dataset that captures the statistical patterns, correlations, and distributions of the original data without containing any real customer information. This proxy data can then replace the original data for downstream use cases such as data sharing across teams, model training, or even public release for research purposes.

Although synthetic data does not directly contain real records, it cannot be assumed to be inherently private. For example, even though the generated data may not directly contain any real user records, the generative model trained on the original data can inadvertently memorize or leak sensitive information about the training inputs. This vulnerability can be exploited through privacy attacks such as membership inference attacks (MIA), where an adversary aims to determine whether a specific individual's data was included in the training set of the generative model [30, 40]. Such privacy leakage is particularly concerning in high-stakes domains like finance, where the data often contains sensitive personal and financial information, such as credit history or transaction records.

To address such vulnerabilities, some works have proposed incorporating differential privacy (DP) into the training process of generative models [13, 39]. DP is a rigorous, mathematical framework that provides provable guarantees against privacy attacks, including MIAs. By introducing carefully calibrated noise during model training, DP ensures that the model's output is not overly influenced by any single individual's data, thus protecting their privacy. While powerful, the application of DP often introduces a fundamental tradeoff between the strength of the privacy guarantee and the downstream utility of the generated synthetic data. A stronger privacy guarantee may require more noise, which can, in turn, degrade the quality and usefulness of the synthetic data [35].

Prior works have investigated the quality, utility, and privacy of different synthetic data generation methods in various domains, such as images [34] and medical data [6, 33]. However, financial datasets have unique characteristics, and as we show, many results from other domains may not translate. A comprehensive understanding of the privacy-utility tradeoff in the financial domain has not yet been fully explored.

Financial datasets often suffer from severe class imbalance, where rare events like loan defaults or fraudulent activities are vastly outnumbered by normal instances. In addition, they typically contain a complex mix of categorical and continuous numerical attributes, which many standard generative models struggle to capture accurately. Finally, the high-dimensional and often sparse nature of financial data further complicates the task of learning the underlying data distribution without overfitting to specific training examples. Such characteristics can exacerbate both privacy leakage and model instability [21], adding to the challenges of generating high-quality synthetic data that is both useful and private.

In this work, we offer a comprehensive empirical study of synthetic data generation for financial datasets. We evaluate a set of representative generative models, Gaussian Copula [24], Tab-Diff [28], CTGAN [36], and TVAE [36]. Further, we provide new reference implementations of differentially private versions of two of these. By making this code publicly available, we offer the community a rigorous foundation for private tabular synthetic data generation. We sample from commonly used financial datasets and study multiple axes, including quality, utility, and privacy metrics. Our contributions are as follows:

(1) Construction of differentially private versions of CTGAN and TVAE that provide rigorous privacy guarantees.[1]
(2) Evaluation of non-private and private generators for multiple financial datasets, measuring quality, downstream utility, and privacy.
(3) Specific exploration of the impact of class imbalance on different metrics.
(4) Discussion of open challenges and recommendations for private synthetic data generation for financial data.

The remainder of this paper is organized as follows. Section 2 summarizes related work. In Section 3, we summarize the synthetic data generation methods we evaluate. Section 4 explains our privacy-preserving implementations of synthetic data generation. Section 5 presents our benchmarking setup, dataset details, and evaluation metrics. Section 6 provides our evaluation results, and in Section 7, we discuss open challenges and recommendations. Finally, we conclude in Section 8.

## 2 Related Work

*Privacy-preserving tabular data synthesis.* Recent years have seen significant advances in combining deep generative methods with DP for tabular data synthesis, with several surveys documenting these developments [15, 26, 29, 37]. As noted in these surveys, tabular data synthesis poses unique challenges compared to other data modalities such as images or text, due to its heterogeneous nature (mix of categorical and continuous features), as well as complex

dependencies among features. This has led to the development of specialized generative models for tabular data, such as CTGAN [36] and TVAE [36], which have shown promising results in generating high-quality synthetic tabular data. However, effectively applying DP to these models is a non-trivial task [15, 29], with some implementations inadvertently leaking training data statistics. In addition, complex models such as CTGAN can suffer from *mode collapse*, where the generator produces a limited variety of outputs, which can be exacerbated by the noise introduced for DP [15]. We directly address these weaknesses by providing versions of CTGAN and TVAE which protect all privacy leak sites, while still incorporating features of CTGAN which mitigate mode collapse. We further explore downsampling as a mitigation for mode collapse in the case of class imbalance.

*Evaluating synthetic data generators.* A challenging question is how to evaluate the privacy of the generated data. Osorio-Marulanda et al. [23] present a systematic review of privacy mechanisms and metrics used to evaluate synthetic data generation, covering both privacy-preserving techniques and evaluation methodologies. In the healthcare domain specifically, Hyrup et al. [17] provides a study of synthetic tabular data generation for healthcare applications, examining the effectiveness of various privacy-preserving techniques. More generally, Steier et al. [31] surveys commonly used metrics for evaluating the privacy of synthetic data, discussing the strengths and limitations of different approaches. Practical implementations of these metrics have been made available through tools such as SDMetrics [8]. However, a strong performance on such metrics may not necessarily correlate with strong privacy implications in real life. Zhao and Zhang [40] examine different methods for synthesizing image data and evaluate the privacy of the resulting data using membership inference attacks (MIA). They find that commonly used privacy metrics such as simple MIA success rates may not correlate well with actual privacy leakage, highlighting the need for more robust evaluation frameworks. Similarly, Ramesh et al. [27] examines the effectiveness of DP when applied to healthcare data, finding that while DP does indeed reduce the overall success of MIAs, privacy risks can still be present depending on how the model is designed. We take a similar approach to these recent works, with distinct results and observations on the interpretation of these metrics in our unique domain.

## 3 Synthetic Data Generation Methods

We compare the quality, utility and privacy of four representative synthetic data generation schemes:
- **Gaussian Copula:** a classical statistical (non-neural) data generator, which estimates the distribution of the input dataset as a multivariate normal distribution;
- **TabDiff**: a recently proposed generative framework that adapts a diffusion architecture to tabular data;
- **CTGAN**: a generative adversarial network (GAN)-based deep learning method to model the distributions of categorical tabular data; CTGAN requires preprocessing to encode continuous attributes.
- **TVAE**: an adaptation of the variational autoencoder model architecture designed alongside CTGAN.

---

[1]The code to reproduce our results can be found in: https://anonymous.4open.science/r/ppsdg-supplementary-3E0D

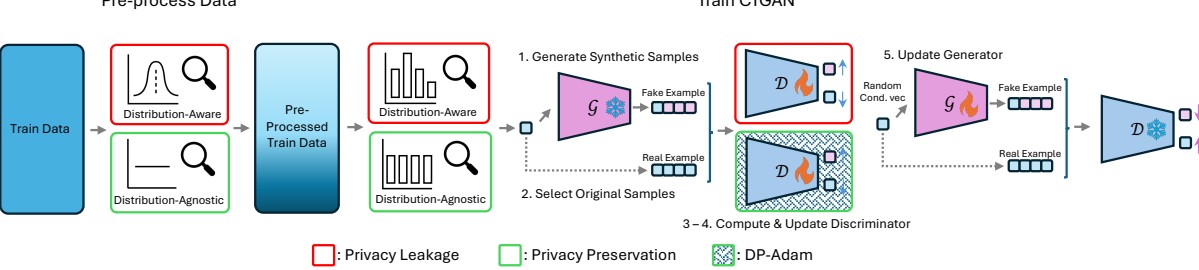

**Figure 1: A visual description of our DP-CTGAN framework, contrasted with the original CTGAN. The privacy leak sites of CTGAN are shown in red boxes, with the privacy-preserving modifications in DP-CTGAN shown in the corresponding green boxes below.**

Additionally, we construct and evaluate DP implementations of CTGAN and TVAE (see Section 4), which we refer to as DP-CTGAN and DP-TVAE, respectively. We select these methods for privacy-protection as they have been shown to generate high quality synthetic data while having algorithmic structures that are amenable to privatization. While DP versions of CTGAN and TVAE have been previously described in the literature [13], these implementations retain features of the original algorithms which represent privacy leakage outside of what is formally accounted for. Using these implementations as a starting point, we make adaptations to eliminate these unaccounted privacy leaks.

## 4 DP Synthetic Data Generation

Differential Privacy (DP) offers a framework for formally quantifying the privacy risk of releasing the results of computation over private data. The intent of DP is to provide protection at the level of individual records, giving some degree of indistinguishability of whether a particular record was used in the production of some output. Formally, we express privacy using the standard definition of $(\epsilon, \delta)$-DP [9].

*Definition 4.1.* A randomized mechanism $\mathcal{M} : \mathcal{D} \to \mathcal{R}$, which takes a dataset from the domain $\mathcal{D}$ as input and produces output of range $\mathcal{R}$, is $(\epsilon, \delta)$-*DP* if, for all datasets $d', d \subset \mathcal{D}$ that differ by inclusion or exclusion of exactly one record and all subsets $S \subset \mathcal{R}$:

$$\Pr(\mathcal{M}(d) \in S) \le \exp(\epsilon) \Pr(\mathcal{M}(d') \in S) + \delta.$$

In the context of synthetic data generation, we consider $\mathcal{M}$ as a generator which takes in real datasets and outputs synthetic datasets of the same form.

### 4.1 DP Implementations

We next detail the steps of DP-CTGAN and DP-TVAE and explain how we guarantee privacy at each step.

*4.1.1 Pre-processing.* CTGAN and TVAE share a mode-specific normalization procedure, an input preprocessing step that bins continuous features into categories according to a Gaussian mixture model. Because this Gaussian mixture is fitted to the input data this binning reveals information about the training data inputs. To address this leak, we replace the Gaussian mixture with a uniform binning transformation based only on the numeric ranges

(min/max) of each column; we assume these ranges are provided as non-private metadata.

*4.1.2 DP-CTGAN.* CTGAN trains a *generator* model to map Gaussian noise to the data distribution alongside an adversarial *discriminator* model to distinguish the generator output from real samples. Below, we describe the training steps of CTGAN, highlight the privacy leak points, and discuss how we address these leaks in DP-CTGAN. The steps are visualized in Figure 1.

*Step 1: Generate synthetic samples.* The generator takes a set of *conditional vectors* as input, each one corresponding to a to-be-generated synthetic sample. The conditional vector specifies a specific column and value for the synthetic sample. CTGAN derives the conditional vectors from aggregate statistics of the input data. In DP-CTGAN, we first use Poisson sampling to select a set of real samples, and then generate each conditional vector using a randomly selected column and the value from a corresponding real sample. The sampling procedure does not expose distributional statistics. The column value is protected in Step 4.

*Step 2: Select original samples.* In CTGAN, a real sample is selected for each conditional vector matching the column value given by the condition. As a result, samples with less common values are selected more often. This mechanism complicates privacy accounting as different samples are used at different rates during training. In DP-CTGAN, we use the real samples from which we generated the conditional vectors. Since this selection is a random process, this simplifies the privacy accounting. The samples themselves are protected in a later step.

*Step 3: Compute discriminator loss and gradient update.* By passing the samples to the discriminator, we generate a prediction of whether each sample is real or synthetic. In CTGAN, we compute the Wasserstein loss on the discriminator by subtracting the mean discriminator output for real samples from the mean discriminator output for synthetic samples. We then add a penalty term computed by taking the mean square distance from 1 of each gradient norm generated by applying the discriminator to a point interpolated uniformly at random along the line between each pair of real and synthetic samples. This penalty term is load-bearing to stabilizing the GAN's performance, but poses a problem for privacy accounting as it mixes information between real samples within the batch. In DP-CTGAN, we accumulate the discriminator loss for each real sample, the synthetic sample generated from its conditional vector,

**Table 1: Details on the evaluation datasets.**

| Dataset | Num. Samples | Num. Features | % Categorical Features | % Minority Class |
|---------|--------------|---------------|------------------------|------------------|
| AD | 48,842 | 14 | 57.14 | 23.9 |
| BC | 10,000 | 11 | 45.45 | 20.3 |
| BM | 45,211 | 13 | 57.14 | 11.7 |
| CC | 30,000 | 23 | 39.13 | 22.12 |
| CR | 1,000 | 20 | 85.00 | 30.00 |
| GM | 150,000 | 10 | 40.00 | 6.68 |

and the gradient penalty computed from their interpolate, and apply clipping to the sum. This isolates the information derived from each single sample to its own sum, which can then be protected with DP noise.

*Step 4: Update the discriminator.* CTGAN updates the discriminator using Adam, which does not protect privacy. We add privacy protection for the training data to the Adam gradient update step by first clipping the gradients on a per-sample basis as identified in the previous step, then adding Gaussian noise as in DP-Adam [1, 19].

*Step 5: Update the generator.* We generate a new batch of synthetic data and compute a generator loss by taking the cross-entropy loss with respect to the condition vectors, which promotes matching the real data, and subtracting the mean discriminator score, which promotes "fooling" the discriminator. The generator is trained using ordinary (non-DP) Adam updates. Because the generator model never accesses the private inputs directly, only the discriminator output, the post-processing property of DP ensures there is no further privacy loss when training the generator, nor when generating synthetic data with it.

*4.1.3 DP-TVAE.* Unlike CTGAN, both the encoder and decoder of TVAE are updated with respect to the original data. Accordingly, we add DP across the entire model by training using DP-Adam.

# 5 Evaluation

## 5.1 Datasets

We use a range of financial datasets from the TabArena benchmark suite [11]. These datasets are all aimed at classification tasks. Each has a mix of categorical and numerical features, and they all exhibit some amount of class imbalance.

- **Adult/Census Income (AD) [4]**: demographic and occupational data from 1994 census database, with label 1 if customer income over $50K/yr and 0 otherwise.
- **Bank Customer Churn (BC) [32]**: customer records, with label 1 if the customer has left the bank and 0 if they did not.
- **Bank Marketing (BM) [22]**: clients from a Portuguese banking institution's phone marketing campaign data, with label 1 if a client subscribes to a term deposit and 0 otherwise.
- **Default of Credit Card Clients (CC) [38]**: credit card payment data from Taiwan, with label 1 if a client defaults in the next month and 0 otherwise.
- **German Credit Data (CR) [16]**: demographic and financial records of German borrowers, with label 1 if the customer is classified as a good credit risk and 2 for a bad credit risk.

- **Give Me Some Credit (GM) [14]**: credit scoring and financial metrics data for borrowers, with label 1 if the borrower experienced financial distress in the next two years and 0 otherwise.

Additional details about the datasets are given in Table 1.

## 5.2 Generator Training & Inference

Appendix A.1 describes the details of the training process for Non-DP generators. For DP-CTGAN and DP-TVAE, we train for 300 epochs using DP-Adam with learning rate 1e-3, $\delta$=1e-5, and clipping norm 1.0. For each dataset, we train with $\epsilon = 1, 5, 10$, giving a range of high, medium, and low privacy protection. We also train with no privacy noise (Adam) to explore the impact of our other adaptations on these methods. We denote these runs by $\epsilon = \infty$ in our results.

For most metrics, we train models 3 times using same training splits but with different random seeding (which varies sampling order and, in DP cases, added noise), and present the average (mean) of each metric across runs under each condition. For the class-balanced subset experiments, we perform a single training run for each condition. For the shadow model attack, we train ten training pairs and one test pair of shadow models for each condition.

## 5.3 Evaluation Metrics

We evaluate each synthetic data generation scheme applied to each selected dataset on metrics expressing both the quality and utility of the generated data, and the privacy loss to the data used to train the generators.

*Quality metrics.* We measure the quality of the synthetic data using two metrics from the SDMetrics [8]. *Column shapes* quantifies the similarity of the distribution of each feature between the original and synthetic datasets. *Column pair trends* quantifies the similarity of correlations between pairs of columns in the two datasets. Both metrics range from 0 to 1, with larger values indicating higher similarity. For each dataset, we report the average column shape and pair trends scores value over all features. We also measure the percentage of samples from the minority class to explore how well the synthetic datasets match the original class distributions.

*Downstream utility.* We measure the utility of each synthetic dataset by the classification performance of an XGBoost model trained on the synthetic data on the test split of the original data. To measure the *best possible* performance on each dataset, we apply hyperparameter optimization (see Appendix A for details). To capture the true performance of the downstream classifier in imbalanced classes, we use the balanced accuracy score, defined as $\frac{1}{2}\left(\frac{TP}{TP+FN} + \frac{TN}{TN+FP}\right)$, where $TP$, $TN$, $FP$, and $FN$ are the numbers of true positives, true negatives, false positives, and false negatives, respectively.

*Privacy metrics.* We use two metrics from SDMetrics [8] to quantify privacy. While these properties do not measure formal DP, they give some indication of whether the synthetic data too closely resembles the training data. The first metric is *distance to closest record (DCR) baseline protection*, which is given by $\min\left(1, \frac{m_{\text{syn}}}{m_{\text{ran}}}\right)$, where $m_{\text{syn}}$ is the median distance between synthetic samples and original samples and $m_{\text{ran}}$ is the median distance between uniformly random data and original; a larger value indicates more privacy. The second metric, *DCR overfitting protection*, scores how much the

**Table 2: Column shapes score for all datasets and generators. Higher values mean more similarity between original and synthetic data.**

| Dataset | Non-DP | | | | DP-CTGAN | | | | DP-TVAE | | | |
|---|---|---|---|---|---|---|---|---|---|---|---|---|
| | Gauss. | TabDiff | CTGAN | TVAE | $\epsilon = 1$ | $\epsilon = 5$ | $\epsilon = 10$ | $\epsilon = \infty$ | $\epsilon = 1$ | $\epsilon = 5$ | $\epsilon = 10$ | $\epsilon = \infty$ |
| AD | 0.885 | 0.972 | 0.855 | 0.862 | 0.710 | 0.696 | 0.716 | 0.759 | 0.692 | 0.626 | 0.595 | 0.756 |
| BC | 0.924 | 0.987 | 0.880 | 0.795 | 0.700 | 0.786 | 0.781 | 0.790 | 0.713 | 0.681 | 0.672 | 0.719 |
| BM | 0.912 | 0.987 | 0.926 | 0.884 | 0.762 | 0.784 | 0.791 | 0.792 | 0.827 | 0.786 | 0.766 | 0.828 |
| CC | 0.873 | 0.980 | 0.847 | 0.903 | 0.640 | 0.650 | 0.657 | 0.669 | 0.702 | 0.618 | 0.633 | 0.867 |
| CR | 0.928 | 0.965 | 0.923 | 0.755 | 0.514 | 0.601 | 0.693 | 0.925 | 0.665 | 0.734 | 0.712 | 0.678 |
| GM | 0.913 | 0.995 | 0.767 | 0.932 | 0.698 | 0.684 | 0.686 | 0.689 | 0.642 | 0.618 | 0.640 | 0.695 |

**Table 3: Column pair trends for all datasets and generators. Higher values indicate more similarity of pairwise column correlations between original and synthetic distributions.**

| Dataset | Non-DP | | | | DP-CTGAN | | | | DP-TVAE | | | |
|---|---|---|---|---|---|---|---|---|---|---|---|---|
| | Gauss. | TabDiff | CTGAN | TVAE | $\epsilon = 1$ | $\epsilon = 5$ | $\epsilon = 10$ | $\epsilon = \infty$ | $\epsilon = 1$ | $\epsilon = 5$ | $\epsilon = 10$ | $\epsilon = \infty$ |
| AD | 0.825 | 0.916 | 0.845 | 0.874 | 0.720 | 0.722 | 0.754 | 0.825 | 0.611 | 0.362 | 0.322 | 0.557 |
| BC | 0.901 | 0.976 | 0.853 | 0.665 | 0.521 | 0.639 | 0.636 | 0.683 | 0.638 | 0.546 | 0.542 | 0.556 |
| BM | 0.847 | 0.983 | 0.813 | 0.808 | 0.844 | 0.889 | 0.854 | 0.846 | 0.582 | 0.543 | 0.491 | 0.723 |
| CC | 0.823 | 0.972 | 0.801 | 0.844 | 0.818 | 0.836 | 0.861 | 0.863 | 0.541 | 0.513 | 0.517 | 0.783 |
| CR | 0.824 | 0.929 | 0.860 | 0.613 | 0.281 | 0.373 | 0.483 | 0.799 | 0.498 | 0.557 | 0.490 | 0.452 |
| GM | 0.881 | 0.991 | 0.642 | 0.770 | 0.916 | 0.892 | 0.899 | 0.871 | 0.701 | 0.611 | 0.571 | 0.631 |

**Table 4: Percentage of the minority class in synthetic data for each dataset and generator, compared to real dataset as baseline. We observe systematic directional skew in minority class representation, with DP-TVAE, in particular, prone to mode collapse.**

| Dataset | Original | Non-DP | | | | DP-CTGAN | | | | DP-TVAE | | | |
|---|---|---|---|---|---|---|---|---|---|---|---|---|---|
| | | Gauss. | TabDiff | CTGAN | TVAE | $\epsilon = 1$ | $\epsilon = 5$ | $\epsilon = 10$ | $\epsilon = \infty$ | $\epsilon = 1$ | $\epsilon = 5$ | $\epsilon = 10$ | $\epsilon = \infty$ |
| AD | 23.9 | 24.1 | 22.8 | 25.3 | 22.7 | 19.6 | 19.6 | 18.7 | 26.3 | 3.09 | 0.04 | 0.00 | 11.8 |
| BC | 20.3 | 21.0 | 21.7 | 30.5 | 13.4 | 19.7 | 20.7 | 18.1 | 20.1 | 0.34 | 0 | 0 | 14.6 |
| BM | 11.6 | 11.7 | 11.2 | 15.1 | 17.0 | 10.8 | 14.3 | 11.5 | 12.3 | 0.37 | 0.01 | 0.00 | 0.23 |
| CC | 22.1 | 21.9 | 20.3 | 32.1 | 14.6 | 19.8 | 18.4 | 14.9 | 27.6 | 1.35 | 0.00 | 0.00 | 8.94 |
| CR | 30.0 | 30.2 | 27.1 | 30.8 | 12.7 | 32.9 | 3.54 | 19.0 | 29.3 | 40.2 | 25.8 | 6.58 | 3.32 |
| GM | 6.68 | 6.68 | 6.32 | 33.0 | 1.86 | 4.14 | 10.4 | 8.96 | 11.9 | 0.64 | 0.02 | 0.15 | 2.75 |

**Table 5: Balanced accuracy of downstream task model trained on the specified dataset.**

| Dataset | Original | Non-DP | | | | DP-CTGAN | | | | DP-TVAE | | | |
|---|---|---|---|---|---|---|---|---|---|---|---|---|---|
| | | Gauss. | TabDiff | CTGAN | TVAE | $\epsilon = 1$ | $\epsilon = 5$ | $\epsilon = 10$ | $\epsilon = \infty$ | $\epsilon = 1$ | $\epsilon = 5$ | $\epsilon = 10$ | $\epsilon = \infty$ |
| AD | 0.818 | 0.516 | 0.684 | 0.718 | 0.769 | 0.681 | 0.709 | 0.734 | 0.784 | 0.508 | 0.5 | 0.5 | 0.753 |
| BC | 0.721 | 0.539 | 0.707 | 0.670 | 0.650 | 0.523 | 0.556 | 0.562 | 0.648 | 0.499 | — | — | 0.619 |
| BM | 0.596 | 0.506 | 0.576 | 0.586 | 0.603 | 0.528 | 0.574 | 0.579 | 0.580 | 0.499 | 0.5 | 0.5 | 0.500 |
| CC | 0.658 | 0.546 | 0.642 | 0.609 | 0.635 | 0.554 | 0.614 | 0.613 | 0.673 | 0.499 | 0.5 | 0.5 | 0.635 |
| CR | 0.686 | 0.533 | 0.614 | 0.507 | 0.552 | 0.501 | 0.494 | 0.505 | 0.555 | 0.518 | 0.515 | 0.524 | 0.553 |
| GM | 0.594 | 0.502 | 0.583 | 0.628 | 0.666 | 0.541 | 0.633 | 0.591 | 0.583 | 0.509 | 0.5 | 0.593 | 0.543 |

synthetic data is overfit to the training data by comparing whether the synthetic data is closer to the original data or a validation set of (non-training) data. This score ranges from 0 to 1, with a higher score indicating less overfitting to the training set.

*Membership inference attack.* We implement a variation on the shadow model attack of Stadler et al. [30]. For a given synthetic data generator method, we first construct a *canary* record, $m$, by selecting a sample arbitrarily from the training data to set aside, and we assign it an incorrect label. From the remainder of the training dataset, we draw the attacker's reference set $\mathcal{R}$ half the size of the training set, and train 10 pairs of shadow models. For each shadow

model pair, we sample a subset $R \subset \mathcal{R}$ of 1/5 the training set size, and train one model on $R$ and the other on $R \cup \{m\}$. Next, we construct a discriminator to classify synthetic datasets sampled from shadow models by whether they were sampled from a model trained with or without the $m$. Details on the discriminator construction are given in Appendix A. To evaluate the attack, we train another pair of test models with the same parameters as the shadow models, with and without the canary, but with their non-canary training examples drawn from the training set instead of the attacker's reference set. The attack success rate is the discriminator's accuracy on a test set of 100 synthetic datasets with 1000 records each sampled randomly

from either test model with equal probability. This represents how well an attacker can distinguish which training set was used in the data generation algorithm, indicating that this algorithm is leaking information about the training set.

For each dataset, we average the results of 10 discriminators on the shadow models, giving the proportion of sampled datasets which were correctly identified by a distinguisher.

## 6 Evaluation Results

### 6.1 Quality Metrics

Table 2 presents the column shapes metric. Among non-DP generators, TabDiff exhibits the best distribution matching across the board, scoring >0.96 on each dataset. The Gaussian Copula model does quite well on this metric, outperforming both CTGAN and TVAE overall; this is not surprising, as the modeling assumptions of the copula model are well-aligned with the metric. We observe significant degradation in column distribution matching in the DP versions of CTGAN and TVAE. Surprisingly, there does not seem to be a strong or consistent correlation between the privacy budget ($\epsilon$) and the column shapes metric. The distributions of individual columns do not seem to be strongly perturbed by the addition of DP noise; the differences between the DP generators and their non-DP counterparts are best attributed to changes in components such as the mode-specific normalization, which cannot be implemented precisely in the private setting.

Table 3 displays the column pair trends metric. Column pair trends require the generators to encode more distributional information about the original data than the column shapes metric, and this increased difficulty is reflected in lower scores across all settings. However, broad trends are comparable between this metric and column shapes. Among non-DP generators, TabDiff again maintains the best distribution matching over all datasets, followed by Gaussian, while CTGAN and TVAE alternately outperform on the same datasets. Between the DP generators, DP-TVAE underperforms DP-CTGAN, which both generally underperform all non-DP generators, which is not unexpected, due to the DP noise in training. As with column shape trends, there is no clear correlation between noise level and pair trends.

Table 4 shows the percentages of samples with the minority target class label in each original (real) dataset and corresponding synthetic datasets. We expect high-quality synthetic data to closely match the class balance of the original dataset. The Gaussian Copula model closely matches the original class balance on every dataset. This is absolutely expected; the Gaussian Copula model expressly encodes the class ratio of a binary column as the mean of that column. TabDiff performs relatively well on this metric, matching each dataset to within 10%, though with a tendency to underrepresent the minority class. CTGAN and TVAE do a very poor job of matching target class distributions. Like TabDiff, TVAE consistently underrepresents the minority class, often to under half of the original frequency. This pattern reflects the typical bias of generative models trained directly on imbalanced classes. CTGAN, in contrast, tends to overrepresent the minority class. This is expected from CTGAN's training-by-sampling mechanism, but suggests that the conditional generator does not compensate accurately for oversampling of the minority label in the target attribute. Moving to the

DP versions, DP-CTGAN outputs synthetic data which approximates the real data's class balance reasonably well and consistently reproduces the original distributions more closely than CTGAN, indicating a robustness to added noise, albeit with increased run-to-run variance under stronger privacy bounds. DP-TVAE outputs are noticeably distorted, with a strong bias to underrepresenting the minority class even without added noise, and in several cases responds to even moderate noise with severe mode collapse.

### 6.2 Downstream Utility

Table 5 shows the balanced accuracy of an XGBoost classifier trained on the original dataset and synthetic datasets using the set-aside test set. Entries marked with "—" indicate that the synthetic dataset contained all samples of a single class. In this case, downstream classification was not attempted.

Among non-DP generators, the Gaussian Copula model reliably yields the lowest balanced accuracy, consistently scoring only slightly above 50%. Despite scoring well on the column shapes, pair trends, and class balance metrics above, these results indicates that the Gaussian Copula model may be poorly matched to capturing categorical relationships that an XGBoost classifier can exploit. The other non-DP methods perform similarly on most datasets and, for the most part, slightly worse than training directly on the original training set. Curiously, however, synthetic datasets generated by both CTGAN and TVAE outperform the original training set on GM, the largest and most imbalanced dataset. While CTGAN's overperformance alone could be attributable to extreme oversampling of the minority class (from 6.68% to 33.0%) creating an artificially balanced dataset, this explanation does not apply to TVAE, which performs even better (66.6% vs CTGAN's 62.8%) despite undersampling the minority class down even further to 1.86%. It is possible that the GM dataset has features which are more easily learned in the output space of generated data than from the real dataset.

Looking to the DP methods, we find that DP-CTGAN without added noise ($\epsilon = \infty$) performs comparably to CTGAN, indicating that our use of privacy-preserving components substantially maintains model utility, consistent with the results of our ablation study in Appendix B.1. As shown in Figure 2, accuracy degrades smoothly as we reduce the privacy budget, showing a clear tradeoff between privacy protection and downstream task performance. This does not clearly result from any consistent shift in class balance, though the variance in class balance increases as noise is added. DP-TVAE performance appears bimodal: in several cases, performance is comparable to TVAE; however, when DP-TVAE exhibits mode collapse, no utility can be extracted from the generated dataset.

### 6.3 Privacy Metrics

Table 6 shows the DCR baseline protection metric. Entries of 0.000 indicate a value smaller than 0.001. Among non-DP generators, the Gaussian Copula consistently scores the highest. TabDiff and CTGAN score similarly, while TVAE shows the lowest DCR baseline values, indicating potentially weaker privacy that is not necessarily compensated for in utility. DP-CTGAN and DP-TVAE score similarly to their non-DP counterparts, with a weak trend toward higher scores as $\epsilon$ decreases.

**Table 6: DCR baseline protection metric. Higher values indicate larger distances to synthetic data vs uniform random baseline.**

| Dataset | Non-DP | | | | DP-CTGAN | | | | DP-TVAE | | | |
|---|---|---|---|---|---|---|---|---|---|---|---|---|
| | Gauss. | TabDiff | CTGAN | TVAE | $\epsilon = 1$ | $\epsilon = 5$ | $\epsilon = 10$ | $\epsilon = \infty$ | $\epsilon = 1$ | $\epsilon = 5$ | $\epsilon = 10$ | $\epsilon = \infty$ |
| AD | 0.321 | 0.246 | 0.102 | 0.020 | 0.212 | 0.175 | 0.111 | 0.070 | 0.186 | 0.005 | 0.003 | 0.006 |
| BC | 0.511 | 0.294 | 0.347 | 0.177 | 0.427 | 0.383 | 0.401 | 0.386 | 0.227 | 0.200 | 0.204 | 0.195 |
| BM | 0.021 | 0.008 | 0.014 | 0.003 | 0.052 | 0.052 | 0.048 | 0.049 | 0.004 | 0.001 | 0.001 | 0.000 |
| CC | 0.141 | 0.017 | 0.086 | 0.013 | 0.149 | 0.079 | 0.066 | 0.058 | 0.013 | 0.009 | 0.009 | 0.011 |
| CR | 0.601 | 0.296 | 0.539 | 0.249 | 0.699 | 0.584 | 0.522 | 0.489 | 0.733 | 0.419 | 0.323 | 0.212 |
| GM | 0.000 | 0.000 | 0.007 | 0.001 | 0.035 | 0.037 | 0.038 | 0.036 | 0.031 | 0.031 | 0.031 | 0.033 |

**Table 7: DCR overfitting protection metric. Higher values indicate that the synthetic data is less overfitted to the original data.**

| Dataset | Non-DP | | | | DP-CTGAN | | | | DP-TVAE | | | |
|---|---|---|---|---|---|---|---|---|---|---|---|---|
| | Gauss. | TabDiff | CTGAN | TVAE | $\epsilon = 1$ | $\epsilon = 5$ | $\epsilon = 10$ | $\epsilon = \infty$ | $\epsilon = 1$ | $\epsilon = 5$ | $\epsilon = 10$ | $\epsilon = \infty$ |
| AD | 0.882 | 0.872 | 0.875 | 0.890 | 0.893 | 0.901 | 0.842 | 0.875 | 0.861 | 0.838 | 0.774 | 0.866 |
| BC | 0.921 | 0.934 | 0.965 | 0.703 | 0.940 | 0.927 | 0.942 | 0.909 | 0.901 | 0.817 | 0.804 | 0.846 |
| BM | 0.869 | 0.877 | 0.882 | 0.908 | 0.751 | 0.760 | 0.750 | 0.756 | 0.880 | 0.867 | 0.851 | 0.920 |
| CC | 0.871 | 0.898 | 0.891 | 0.861 | 0.760 | 0.760 | 0.760 | 0.750 | 0.855 | 0.854 | 0.837 | 0.861 |
| CR | 0.918 | 0.777 | 0.914 | 0.853 | 0.615 | 0.733 | 0.762 | 0.908 | 0.856 | 0.862 | 0.918 | 0.728 |
| GM | 0.614 | 0.652 | 0.736 | 0.565 | 0.496 | 0.500 | 0.454 | 0.448 | 0.529 | 0.535 | 0.532 | 0.438 |

**Table 8: Membership inference attack success rate. Higher values indicate more successful identification of whether a synthetic dataset was output by a generator trained with a specified canary, with 0.5 equivalent to random guessing.**

| Dataset | Non-DP | | | | DP-CTGAN | | | | DP-TVAE | | | |
|---|---|---|---|---|---|---|---|---|---|---|---|---|
| | Gauss. | TabDiff | CTGAN | TVAE | $\epsilon = 1$ | $\epsilon = 5$ | $\epsilon = 10$ | $\epsilon = \infty$ | $\epsilon = 1$ | $\epsilon = 5$ | $\epsilon = 10$ | $\epsilon = \infty$ |
| AD | 0.491 | 0.507 | 0.505 | 0.496 | 0.513 | 0.497 | 0.512 | 0.475 | 0.509 | 0.484 | 0.502 | 0.508 |
| BC | 0.536 | 0.49 | 0.426 | 0.501 | 0.504 | 0.499 | 0.499 | 0.514 | 0.474 | 0.495 | 0.529 | 0.483 |
| BM | 0.523 | 0.482 | 0.505 | 0.515 | 0.491 | 0.487 | 0.49 | 0.488 | 0.51 | 0.519 | 0.494 | 0.507 |
| CC | 0.498 | 0.549 | 0.412 | 0.489 | 0.494 | 0.499 | 0.471 | 0.494 | 0.5 | 0.524 | 0.518 | 0.491 |
| CR | 0.530 | 0.502 | 0.497 | 0.488 | 0.517 | 0.495 | 0.516 | 0.485 | 0.518 | 0.476 | 0.513 | 0.519 |
| GM | 0.48 | 0.528 | 0.453 | 0.522 | 0.532 | 0.516 | 0.477 | 0.474 | 0.465 | 0.479 | 0.494 | 0.478 |

**Table 9: Balanced accuracy difference between models trained from downsampled vs original dataset. Positive values indicate higher accuracy from model trained with downsampled data.**

| Dataset | Non-DP | | | | DP-CTGAN | | | | DP-TVAE | | | |
|---|---|---|---|---|---|---|---|---|---|---|---|---|
| | Gauss. | TabDiff | CTGAN | TVAE | $\epsilon = 1$ | $\epsilon = 5$ | $\epsilon = 10$ | $\epsilon = \infty$ | $\epsilon = 1$ | $\epsilon = 5$ | $\epsilon = 10$ | $\epsilon = \infty$ |
| AD | +47.5% | +18.6% | +12.5% | +6.2% | -3.4% | -1.8% | -4.1% | -1.9% | +2.6% | +4.4% | +3.4% | +5.3% |
| BC | +29.5% | -29.6% | +3.9% | +9.1% | -4.0% | +2.3% | -0.7% | +0.6% | +6.0% | +14.2% | +22.6% | +7.8% |
| BM | +26.1% | +17.0% | +14.3% | +11.9% | -1.1% | -4.5% | +5.2% | -2.6% | +6.6% | +8.2% | +8.4% | +29.2% |
| CC | +28.4% | +11.1% | +14.0% | +11.0% | -3.8% | +3.9% | -4.4% | -5.3% | +5.4% | +12.8% | +8.4% | +11.2% |
| CR | +8.8% | -10.9% | -13.2% | +10.7% | -2.6% | -0.4% | +3.6% | +2.0% | -1.9% | -1.6% | -1.0% | +10.7% |
| GM | +49.6% | +34.1% | +20.2% | +14.7% | -5.4% | +4.3% | +2.0% | -3.8% | +3.1% | +20.8% | +7.3% | +39.4% |

We observe that these scores are much more strongly real dataset dependent than they are sensitive to qualities of the generator, with most scores on the same dataset lying within an order of magnitude across all generators, while varying wildly between datasets even when the same generator is used. The scale of the baseline protection score is most influenced by the (non-)uniformity of the data distribution of the real dataset, as the uniform random baseline places points randomly within the ranges of all attributes, whereas we expect synthetic data to proportionately represent clusters in the training data.

Table 7 shows the DCR overfitting protection metric. Like the DCR baseline metrics, these seem to be more driven by features of

the dataset than the generator, with the GM dataset in particular scoring relatively low on both metrics over all generators. While the dataset-based score trend holds for DP-CTGAN and DP-TVAE at all privacy budgets, we observe no apparent relation between $\epsilon$ and the DCR overfitting score. Our intuition is that that both of these metrics may be too sensitive to dataset-specific attributes to serve as general privacy evaluation metrics.

## 6.4 Membership Inference Attack Results

Table 8 presents success rates of the shadow model membership inference attack. We immediately see that most of the attack success rates are close 0.5, i.e., equivalent to random guessing. We

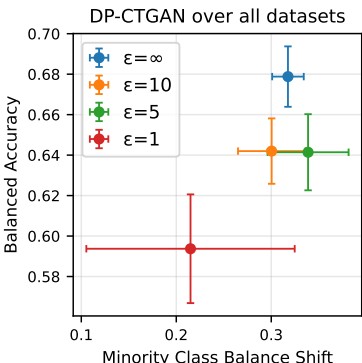

**Figure 2: Comparison of the shift (absolute difference) in minority class percentage of the synthetic dataset over original, and resulting balanced accuracy of a downstream model trained on the synthetic dataset. Each plot represents mean and standard deviation over all datasets. As $\epsilon$ increases (and privacy decreases), we see a clear increase in accuracy and decrease in the variance of minority class balance shift.**

do observe some attack success rates further from 0.5 for specific generator-dataset pairs. The direction of these outliers are roughly evenly split, and there is no clear pattern to the conditions where they appear, so it is difficult to conclude from these results whether any method yields better privacy than another. In fact, we find that the most common outcome is that the discriminator outputs the same prediction for all inputs. Interestingly, however, the variance over all MIA success rates presented is significantly higher than expected ($\sigma = 0.0218$ vs expected $\sigma = 0.0158$) for the corresponding binomial distribution with 1000 trials and success rate 0.5.

Overall, these results show that, in the case of financial datasets, the MIA success rate is not a reliable proxy for the theoretical guarantees provided by these DP generators, and that more research is needed into privacy auditing in domain-specific settings.

### 6.5 Impact of Balancing Classes

To isolate the impact of training data class imbalance on the utility of generated data, we evaluate the generators on balanced datasets. We generate balanced subsets of each dataset by downsampling the majority class in each training set to match the size of the minority class, then train each generator on the resulting smaller, balanced dataset. Results in this section represent a single training run under each experimental condition.

We find that all generators maintain close to the 50/50 class balance of the input as expected (see Table 12 in the Appendix). This effectively eliminates mode collapse, though the resulting output distributions are obviously no longer representative of the full dataset. Table 9 shows the percentage difference in downstream balanced accuracy between XGBoost classifiers trained using synthetic data generated on downsampled datasets versus full datasets.

Most generators show modest improvements on each dataset compared to its imbalanced counterpart. We expect some improvement, as better class balance improves uptake of class-specific distributional differences in both the synthesizer and the task model.

Among the non-DP generators, we specifically see the largest improvements from the Gaussian Copula generator, with a consistent but less pronounced effect on TVAE. CTGAN and TabDiff perform overall but not consistently better with balanced inputs. The overall trend is that models which already performed well on the downstream task are less improved by providing balanced input.

## 7 Discussion

*Privacy auditing challenges.* Our evaluation indicates that there are difficulties in empirically quantifying privacy in synthetic data generation. Existing metrics poorly quantify privacy in a way that can be interpreted consistently across datasets or even privacy levels. Further, there is not a strong correlation between DCR metrics and the MIA success rate, raising questions about which metrics truly capture privacy. More work is needed on defining privacy, constructing privacy attacks tailored for this setting, and integrating those attacks into a rigorous mathematical framework that can give formal privacy bounds.

*Comparison with other studies.* Compared to recent work on image data [40], we observe a much weaker ability to identify privacy violations through attacks on generated data. This aligns our results with prior work on evaluating privacy risk in medical data [2, 6], which also identify limitations in MIA. This leads us to believe that we *cannot* assume that low MIA success rates imply that private data is well-protected by synthetic data generation. Further, it indicates the importance of evaluating privacy and privacy metrics in a variety of domains.

*Generalization to other domains.* We believe that the defining challenges we encounter in data generation on financial datasets, with mixed numerical and categorical features with severe class imbalance, are also generalizable to other classes of sensitive or regulated settings. For example, medical datasets often present the same pattern where a rare class of record (e.g., specific disease diagnosis) is both of interest and highly sensitive from a privacy perspective. Future work is needed to determine which characteristics are most relevant in determining the suitability of particular generation schemes or privacy auditing method.

## 8 Conclusion

We have studied privacy-utility tradeoffs among synthetic data generation schemes on tabular financial datasets. We assessed the quality and utility of generated data using column trend metrics and downstream task performance, finding that while we are able to adapt state-of-the-art synthetic data generators to data distributions with good utility with the option to preserve privacy, implementing rigorous privacy protection comes with noticeable tradeoffs on downstream task performance. Our results indicate a need for domain-specific privacy-preserving synthetic data generators, which is a subject for future work.

## Acknowledgment

Authors acknowledge the support from NSF IUCRC CRAFT center research grant (CRAFT Grant #22023) for this research. The opinions expressed in this publication do not necessarily represent the views of NSF IUCRC CRAFT.

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

**Table 10: Hyperparameters used for tuning downstream XGBoost classifier.**

| Name | Values |
|---|---|
| n_estimators | randint(100,1000) |
| max_depth | randint(3,10) |
| learning_rate | loguniform(0.005,0.01) |
| min_child_weight | randint(1, 5) |
| colsample_bytree | uniform(0.5, 1.0) |

# A Implementation Details

## A.1 Generator Training Details

We train our generators using a fixed random seed. Each dataset is split into training, validation and test sets at a ratio of 8:1:1. The splits are generated with respect to the target class balance to ensure that the class distribution is consistent across the splits. The synthetic data generators are trained only using the train split of the original data, ensuring that we have held-out splits from the original data that we can later use to validate the downstream utility of the synthetic data. We use scikit-learn [25] to pre-process and split our datasets.

**Table 11: Balanced accuracy on the downstream task model on trained against ablations of CTGAN. Results are given as the average over three generator training runs, but tested over only a subset of datasets. We identify the WGAN gradient penalty and gradient clipping as influential components.**

| Dataset | Original | Non-DP | | | | | |
|---|---|---|---|---|---|---|---|
| | | CTGAN | UniSamp | BatchSamp | NoPenalty | UniTrans | GradClip |
| AD | 0.818 | 0.718 | 0.729 | 0.728 | 0.547 | 0.729 | 0.684 |
| BC | 0.726 | 0.648 | 0.601 | 0.589 | 0.559 | 0.649 | 0.707 |
| PW | 0.974 | 0.923 | 0.919 | 0.904 | 0.916 | 0.908 | 0.929 |

**Table 12: Minority class percentage for synthetic datasets generated from the downsampled class-balanced original dataset. Results are given for a single synthetic dataset.**

| Dataset | Original | Non-DP | | | | DP-CTGAN | | | | DP-TVAE | | | |
|---|---|---|---|---|---|---|---|---|---|---|---|---|---|
| | | Gauss. | TabDiff | CTGAN | TVAE | $\epsilon = 1$ | $\epsilon = 5$ | $\epsilon = 10$ | $\epsilon = \infty$ | $\epsilon = 1$ | $\epsilon = 5$ | $\epsilon = 10$ | $\epsilon = \infty$ |
| AD* | 50 | 49.2 | 42.7 | 36.1 | 48.7 | 49.6 | 48.0 | 49.2 | 49.9 | 44.8 | 21.7 | 15.6 | 41.5 |
| BC* | 50 | 49.2 | 49.0 | 47.7 | 49.8 | 49.2 | 45.4 | 49.3 | 46.0 | 25.4 | 48.4 | 34.4 | 49.2 |
| BM* | 50 | 49.7 | 45.6 | 44.9 | 43.6 | 48.7 | 49.5 | 49.7 | 49.6 | 45.7 | 47.9 | 40.4 | 40.0 |
| CC* | 50 | 49.8 | 49.1 | 41.9 | 47.5 | 47.2 | 47.2 | 48.9 | 48.5 | 45.6 | 49.9 | 29.9 | 49.6 |
| CR* | 50 | 48.8 | 46.1 | 49.8 | 49.4 | 27.6 | 41.7 | 45.3 | 42.1 | 37.6 | 46.7 | 47.9 | 43.9 |
| GM* | 50 | 49.7 | 44.7 | 37.7 | 48.6 | 48.1 | 49.0 | 48.8 | 49.5 | 41.8 | 42.8 | 49.7 | 48.0 |

**Table 13: Balanced accuracies for downsampled class-balanced original dataset and synthetic datasets generated from the downsampled original dataset. Results are given for a single synthetic dataset.**

| Dataset | Original | Non-DP | | | | DP-CTGAN | | | | DP-TVAE | | | |
|---|---|---|---|---|---|---|---|---|---|---|---|---|---|
| | | Gauss. | TabDiff | CTGAN | TVAE | $\epsilon = 1$ | $\epsilon = 5$ | $\epsilon = 10$ | $\epsilon = \infty$ | $\epsilon = 1$ | $\epsilon = 5$ | $\epsilon = 10$ | $\epsilon = \infty$ |
| AD* | 0.722 | 0.761 | 0.811 | 0.808 | 0.817 | 0.658 | 0.696 | 0.704 | 0.769 | 0.521 | 0.522 | 0.517 | 0.793 |
| BC* | 0.808 | 0.698 | 0.498 | 0.696 | 0.709 | 0.502 | 0.569 | 0.558 | 0.652 | 0.529 | 0.571 | 0.613 | 0.667 |
| BM* | 0.596 | 0.638 | 0.674 | 0.670 | 0.675 | 0.522 | 0.548 | 0.609 | 0.565 | 0.532 | 0.541 | 0.542 | 0.646 |
| CC* | 0.654 | 0.701 | 0.713 | 0.694 | 0.705 | 0.533 | 0.638 | 0.586 | 0.637 | 0.526 | 0.564 | 0.542 | 0.706 |
| CR* | 0.719 | 0.580 | 0.547 | 0.440 | 0.611 | 0.488 | 0.492 | 0.523 | 0.566 | 0.508 | 0.507 | 0.519 | 0.612 |
| GM* | 0.592 | 0.751 | 0.782 | 0.755 | 0.764 | 0.512 | 0.660 | 0.603 | 0.561 | 0.525 | 0.604 | 0.636 | 0.757 |

For CTGAN and TVAE, we train for 300 epochs using Adam with learning rate 1e-3. For TabDiff, we train for 8000 epochs using AdamW with learning rate 1e-3 for full-dataset metrics, taking the model with the lowest training loss occurring after the 4000th epoch.

## A.2 Hyperparameter Optimization for Downstream Classifier Training

We apply hyperparameter optimization on XGBoost [5] classifier using the Optuna [3] algorithm from Ray Tune library [20] for 100 trials. The best set of hyperparameters is picked based on their performance on the validation split to ensure that the test split is never used in any part of training.

We use the the same set of hyperparameters used in the Auto-Gluon [10] library, as shown in Table 10.

## A.3 Details of the MIA Discriminator

We construct a discriminator to classify synthetic datasets sampled from shadow models by whether they were sampled from a model trained with or without the canary. To reduce the dimensionality of the distribution of synthetic datasets, we extract a histogram feature set from each dataset sampled from a shadow model by binning numerical features, using the CTGAN mode-specific normalization

fitted on a reference data to select the bins, and computing the marginal frequencies of each attribute of the processed dataset. Our discriminator is a Random Forest classifier with 100 classifiers using Gini splitting, trained on 100 feature sets extracted from 1000-record synthetic datasets, each sampled from a randomly selected shadow model, labeled with whether that shadow model was trained using the canary. The same feature set is used to preprocess the synthetic datasets sampled from the test models.

## B Additional Experimental Results

### B.1 Generator Ablation Study

We identified several components of CTGAN which were problematic from the perspective of differential privacy accounting, and our construction of DP-CTGAN eliminates or replaces each of these features with counterparts that are more amenable to privacy analysis. We manage to achieve this while substantially maintaining downstream task utility, a surprising result as several of these features are identified as key elements of CTGAN's performance. Table 11 shows downstream task balanced accuracies for a selection of datasets and modifications of CTGAN:

- UniSamp: Condition vectors are sampled according to raw class frequencies, not log frequencies (CTGAN's "training-by-sampling").

- BatchSamp: Instead of selecting real samples based on condition vectors, real batches are sampled directly from the training set at large.
- NoPenalty: Discriminator is trained without the WGAN gradient penalty term.
- UniTrans: Uniform binning replaces GMM transformer.
- GradClip: Discriminator gradients clipped to a max norm (1.0/step).

We observe no substantial effect of training-by-sampling and the mode-specific normalization on downstream performance, while the effect of gradient clipping is inconsistent. However, the WGAN gradient penalty appears to be load-bearing, as its removal induces significant reduction in utility across the board. This observation motivates our redesign of DP-CTGAN to compute a "per-sample" gradient penalty for each real sample.

## B.2 Data for Impact of Balancing Classes

Table 12 shows the frequency of the minority target class label in datasets generated after downsampling to 50/50 class balance. Table 13 shows raw balanced accuracy of a XGBoost classifier trained on the output datasets.

We find that all generators produce synthetic data with substantial representation of both classes when equally represented in the input, eliminating the worst cases of mode collapse where downstream utility was lost, though we do see significant class balance skew in some conditions. Downstream task performance is overall comparable compared to direct use of the imbalanced data, despite use of smaller datasets after downsampling. There is a slight performance increase overall, though the bulk of that comes from improvements on Gaussian Copula, TVAE, and DP-TVAE, the worst-performing models, while effects are more ambivalent for TabDiff, CTGAN, and DP-CTGAN.