# OpenReview forum: "Measuring Privacy Risks and Tradeoffs in Financial Synthetic Data Generation"
_ACM.org/TheWebConf/2026/Workshop/TIME — TIME 2026 Poster_

### Meta-Review · Program_Chairs · 2026-02-02

**Recommendation:** Accept (Poster)
**Confidence:** 4

**Metareview:**

This work is submitted after the workshop deadline; hence the review is processed by program chair.

This paper identifies a key problem in data, and the evaluations and analysis provide some insights regarding the data from different aspects. The authors also provided the code for the review, which is good.

There are some major concerns which PC suggests the authors to thoroughly revise before the paper could be accepted:

- In abstract section, "we provide novel privacy-preserving implementations of two of these" appears to be unclear. It would be better to be more specific.

- In intro, the authors listed 4 contributions; however, the abstract section does not reflect on any of these contributions.

- Related work section does not outline the major differences between this work and existing works in the current literature.

- Be consistent in the whole paper regarding the names. For example, in Line 124, "Copula" but in line 200, it becomes "copula", the authors should use standard names for the method / model names.

- Fig. 1 presents the overview of the framework. However, the figure caption does not provide much information to understand the framework. Also it would be better to have few sentences describe a bit the core innovations and provide some insights.

- Sec 4 starting paragraph is unclear. Some of the maths symbols are not being properly explained before using them, such as what is $e$? Also these concepts need to be properly and clearly explained to ensure readers can understand.

- Another major weakness is that all the table captions do not provide explanations or descriptions. Some tables contain abbreviations; however, these abbreviations are not being properly explained or described. It would be difficult for readers to understand these tables. Also adding few sentences describe the insights / findings would be more beneficial.

- Fig. 2 could be in more detail. It would be better to also explain eg "balanced accuracy", "minority class balance shift". These concepts need to be clear in figure caption.

- PC also reads the appendix, and for the tables presented in the appendix, they also need to be self-explainable, also it would be better to add some descriptions / findings in texts rather than simply refer readers to tables.

- The authors should ensure that all references are validated (not hallucinated).

The authors should revise the paper as suggested, and ensure all author names, affiliations, and email addresses, acknowledgments (if exists) are properly provided as camera ready version. Please submit the camera-ready version on openreview system, it would be better to make the code publicly available to benefit the research community.

---

### Decision · Program_Chairs · 2026-02-03

**Decision:**

Accept (Poster)

**Comment:**

The authors have provided a revision based on meta review comments.